# Ribitol in Solution Is an Equilibrium of Asymmetric Conformations

**DOI:** 10.3390/molecules26185471

**Published:** 2021-09-08

**Authors:** Shiho Ohno, Noriyoshi Manabe, Takumi Yamaguchi, Jun Uzawa, Yoshiki Yamaguchi

**Affiliations:** 1Division of Structural Glycobiology, Institute of Molecular Biomembrane and Glycobiology, Tohoku Medical and Pharmaceutical University, 4-4-1 Komatsushima, Aoba-ku, Sendai 981-8558, Miyagi, Japan; s.ohno@tohoku-mpu.ac.jp (S.O.); manabe@tohoku-mpu.ac.jp (N.M.); 2School of Materials Science, Japan Advanced Institute of Science and Technology, 1-1 Asahidai, Nomi 923-1292, Ishikawa, Japan; takumi@jaist.ac.jp; 3Structural Glycobiology Team, RIKEN (The Institute of Physical and Chemical Research), 2-1 Hirosawa, Wako 351-0198, Saitama, Japan; uzawa_fe79@ck9.so-net.ne.jp

**Keywords:** ribitol, conformation, dynamics, NMR, MD simulation, hydrogen bond

## Abstract

Ribitol (C_5_H_12_O_5_), an acyclic sugar alcohol, is present on mammalian α-dystroglycan as a component of *O*-mannose glycan. In this study, we examine the conformation and dynamics of ribitol by database analysis, experiments, and computational methods. Database analysis reveals that the anti-conformation (180°) is populated at the C3–C4 dihedral angle, while the gauche conformation (±60°) is seen at the C2–C3 dihedral angle. Such conformational asymmetry was born out in a solid-state ^13^C-NMR spectrum of crystalline ribitol, where C1 and C5 signals are unequal. On the other hand, solution ^13^C-NMR has identical chemical shifts for C1 and C5. NMR ^3^*J* coupling constants and OH exchange rates suggest that ribitol is an equilibrium of conformations, under the influence of hydrogen bonds and/or steric hinderance. Molecular dynamics (MD) simulations allowed us to discuss such a chemically symmetric molecule, pinpointing the presence of asymmetric conformations evidenced by the presence of correlations between C2–C3 and C3–C4 dihedral angles. These findings provide a basis for understanding the dynamic structure of ribitol and the function of ribitol-binding enzymes.

## 1. Introduction

Ribitol (C_5_H_12_O_5_) is an acyclic sugar alcohol and a component of teichoic acid found in Gram-positive bacteria [1] (Figure 1). Ribitol is also found in riboflavin (vitamin B2) and flavin mononucleotide (riboflavin 5′-phosphate). Ribitol phosphate (d-ribitol 5-phosphate) is a component of *O*-mannose glycans on mammalian α-dystroglycan [2]. A genetic deficiency of ribitol phosphate transferase leads to muscular dystrophy, highlighting the essential role of ribitol phosphate in the development of skeletal muscle [3]. Laminin is known to interact with ribitol-containing *O*-mannose glycans, and the ribitol residues are likely acting as a hinge in bridging laminin and α-dystroglycan. Hence, the structure and dynamics of ribitol are now receiving much attention. However, compared with the cyclic hemiacetal sugars, our knowledge of the structural properties and dynamics of such an acyclic sugar alcohol is rather limited.

Jeffrey et al. investigated the crystal structures of nine alditols, including ribitol [4]. In the crystalline state, the carbon chain of alditols tends to adopt an extended, planar zigzag conformation when the configurations at alternate carbons are different (e.g., C_n_ and C_n+2_ will be d and l or l and d). When the configurations of alternate carbons are the same (d and d or l and l), the C_n_-O and C_n+2_-O bonds align in parallel. This arrangement causes steric hindrance such that the carbon chain changes its conformation to bent and non-planar. Ribitol adopts the same configuration at C2 and C4 (d and d), and the steric hindrance brought by O2 and O4 is avoided by rotating the C2–C3 or C3–C4 bonds by 120°. In fact, the crystal structure of ribitol adopts a bent conformation, avoiding the steric hindrance induced by a stretched planar zigzag structure. Likewise, the ribitol moiety of riboflavin exhibits a bent conformation in crystals [5,6].

Hawkes et al. performed solution NMR analysis to determine the structure of ribitol based on ^3^*J*(H,H) values [7]. Ribitol equilibrates between a twist at C2–C3 and one at C3–C4, removing the O2/O4 interaction in the planar chain form. It should be noted that ^3^*J*(H2,H3) and ^3^*J*(H3,H4) of ribitol are indistinguishable in solution NMR. The relationship between C2–C3 and C3–C4 torsion angles is not definitive solely from solution NMR analysis.

Garrett and Serianni synthesized ^13^C-labeled sugar alcohols, including D-[1-^13^C] ribitol, for NMR analysis [8]. ^3^*J*(C1,C4) is 1.7 Hz, which is intermediate between ≈1.3 Hz (gauche conformations) and ≈2.3 Hz (anti conformation), indicating that the C2–C3 or C3–C4 linkage is a mixture of extended and bent forms. Franks et al. found that the ^3^*J*(H,H) values are significantly different in water and organic solvents, suggesting that hydration and hydrogen bonding affect the conformation of ribitol [9].

Klein et al. performed MD and Monte Carlo (MC) simulations for the tetrasaccharide-ribitol unit in teichoic acid [9]. They found that the GalNAc-ribitol linkage is more flexible than other glycosidic linkages, but the ribitol chain itself is not completely flexible. Hatcher et al. parameterized the CHARMm force field for acyclic sugar alcohols and performed MD calculations for sugar alcohols, including ribitol [10]. Other than these, there are few studies on the conformational dynamics of ribitol.

Our present study aims to build on previous knowledge on the conformation of ribitol and to deepen our understanding of static and dynamic structures of ribitol through database analysis, experimental NMR analysis, and MD simulation.

## 2. Results and Discussion

### 2.1. Database Analysis of Ribitol and Ribitol Phosphate

Crystal structures of oligosaccharides bound to proteins provide good insight into the more stable conformations of each glycosidic linkage [11]. Following this notion, we extracted the conformation(s) of ribitol and ribitol phosphate from the Cambridge Crystallographic Data Centre (CCDC) and the Protein Data Bank (PDB), in total, six ribitol and four ribitol-phosphate structures (Figure 2).

The dihedral angles of the carbon main chain and the C1–C5 distance of extracted structures are listed in Table 1. Although the data are limited, one can examine the stable conformations for each dihedral angle. In each structure, the dihedral angles suggest staggered conformations. Except for one example (6H4M), the four dihedral angles (φ1–φ4) do not all simultaneously assume anti-conformations; some prefer bent conformations.

When φ2 (C1–C2–C3–C4) is in the anti-conformation, φ3 (C2–C3–C4–C5) adopts gauche and vice versa. These results indicate that φ2 and φ3 correlate with each other. C1–C5 distances were also calculated to ascertain the degree of carbon chain stretch. The C1–C5 distance of the fully extended ribitol measured 5.0 Å. The average value of the C1–C5 distance of 10 extracted ribitol and ribitol phosphate structures is 4.6 Å, suggesting that ribitol and ribitol phosphate are not completely extended, possibly to avoid the continuous anti-conformations that induce O2/O4 repulsion. In high-resolution structures of ribitol (PDB ID; 5IAI and 4Q0S), hydrogen bonding with surrounding water molecules is observed (Figure 2c). This suggests that intermolecular interaction with water molecules may be involved in the various conformations of ribitol. It seems that the phosphorylation of ribitol does not significantly change the ribitol conformation, although it needs further detailed study. In this paper, we mainly focus on ribitol.

### 2.2. NMR Analysis

#### 2.2.1. Solid-State ^13^C-NMR Analysis of Crystalline Ribitol

We can conclude from the database analysis that stable conformations of ribitol are likely asymmetric. To investigate the conformation of crystalline ribitol from another aspect, solid-state ^13^C-NMR was performed. When the conformation of ribitol is asymmetric, carbon signals will be observed separately in the NMR spectra. Indeed, a ^13^C CP-MAS NMR spectrum of crystalline ribitol shows four separate signals (Figure 3a). By comparison with the solution ^13^C-NMR spectrum of ribitol (Figure 3b), partial assignments of ^13^C signals were possible for 72 and 73 ppm to C2/C3/C4 and the peaks at 60 and 62 ppm to C1/C5. Interestingly, C1/C5 shows two separate peaks. This observation may indicate that C1 and C5 of ribitol are in different microenvironments. As seen above, ribitols deposited in CCDC have an asymmetric structure (Table 1). It is plausible that ribitol has an asymmetric structure in the solid state as well, which gives different chemical shifts. C2/C3/C4 signals appear around the same frequency; therefore, interpretation here is impossible without a complete assignment.

In contrast, the solution ^13^C-NMR of ribitol showed that C1 and C5, C2 and C4 have the same chemical shifts (Figure 3b). These results suggest that ribitol transits rapidly in solution between stable asymmetric conformations, giving rise to averaged signals.

#### 2.2.2. Solution NMR Analysis of Ribitol Based on Coupling Constant

Then, the dihedral angle of the ribitol main chain was analyzed from the coupling constants obtained by solution NMR. We initially tried to obtain ^3^*J*(H,H) from the splitting of the signals, assuming a first-order approximation. However, the assumption was not valid due to strong coupling in ribitol, even at the ^1^H observation frequency of 600 MHz. Therefore, ^3^*J* values were estimated by comparing simulated and experimental spectra iteratively. To improve the accuracy, 1D-^1^H NMR measurements were performed at the ^1^H observation frequencies of 270, 400, and 600 MHz, and the iterative comparison was done at each frequency (Figure 4). ^3^*J*(C,H) was estimated from the HR-HMBC method [18]. The observed chemical shifts and coupling constants are summarized in Table 2.

Chemical shifts and ^2,3^*J(*H,H) coupling constants are in substantial agreement with those of previous studies [7,8,10,19]. ^3^*J(*C1,H3) was measured as 3.8 Hz in this study, similar to a previous report (3.7 Hz) [8]. Then, these coupling constants are used for estimating the distribution of each conformer using the Haasnoot equation, which includes a set of empirical constants [20]. Regarding the O1–C1 dihedral angle (φ1 = O1–C1–C2–C3), the population of three conformers (φ1 = 180°, −60°, and +60°) was calculated using ^3^*J*(H1R,H2)(*pro-R*) and ^3^*J*(H1S,H2)(*pro-S*) with reference to the method of Hawkes [7]. The φ1 ratio was estimated as 180°:−60°:+60° = 64:36:0. This closely conforms to the result of Hawkes [7]. For the C1–C2 dihedral angle φ2 (φ2 = C1–C2–C3–C4), the conformations can be estimated from ^3^*J*(H2,H3) and ^3^*J*(C1,H3). The population of each φ2 conformation was calculated as 180°:−60°:+60° = 2:46:52. These data suggest the presence of a rapid transition (>>10 Hz in terms of coupling constant) around the C1–C2 axis and a relatively small population of the anti-conformation. It should be noted that φ1 and φ4 or φ2 and φ3 cannot be discussed separately because their NMR signals (e.g., H1 and H5, H2 and H4) overlap in solution.

#### 2.2.3. NMR Analysis of Hydroxy Protons

Franks et al. found that the ^3^*J*(H,H) values of ribitol differ significantly in water and organic solvents [19], and this suggests that hydration and hydrogen bonding affect the conformation of ribitol. In this study, we analyzed the hydroxyl group of ribitol by directly observing the hydroxyl proton by ^1^H-NMR. The assignments of OH signals are shown in Figure 5. At 0 °C, two sharp peaks are observed at 5.9 ppm and 5.8 ppm, and these correspond to OH3 and OH2/OH4 (Figure 5a). The OH3 signal gives a relatively sharp peak at 0 °C, suggesting that OH3 is stabilized by hydrogen bonding. To validate the exchange of the hydroxyl proton, the temperature coefficient (ppb K^−1^) was estimated as 11 ppb K^−1^ for ribitol OH3. According to Sandströ et al., the temperature coefficient is greater than 11 ppb K^−^^1^ when the OH proton is fully hydrated [8]. This suggests that OH3 is mostly exposed to solvent, but some transient hydrogen bond may exist. The exchange rate constant of OH3 was also estimated from 2D EXSY. From the build-up curve of exchange peak volume, it was estimated as *k*_ex_ = 193 s^−1^. To estimate the effect of an intramolecular hydrogen bond on the temperature coefficient and OH exchange rate, the same experiment was performed using the model compound 1,3,5-pentanol (Figure 1), which does not have hydroxyl groups at the 2 and 4 positions. 1,3,5-Pentanol gives a broad OH peak around 5.7 ppm at 0 °C. This peak can be assigned to OH1/OH3/OH5, and the peak almost disappears at 25 °C (Figure 5b). We reasoned that this is due to the lack of a hydrogen bond, and hence, OH3 shows faster proton exchange.

For comparison, the temperature coefficient of 1,3,5-pentanol was calculated as 13 ppb K^−1^ for OH1/3/5. Similarly, the exchange rate of OH1/3/5 was estimated from EXSY to be *k*_ex_ = 650 s^−1^. These observations suggest that ribitol OH3 tends to form water-mediated or intramolecular hydrogen bonds.

### 2.3. MD Simulations

#### Distribution of Main Chain Dihedral Angles

Since solution NMR does not distinguish between ^3^*J*(H2,H3) and ^3^*J*(H3,H4), MD simulations were performed to shed light on the dihedral angles of φ2 and φ3 separately. We used three ribitol coordinates in PDB (PDB ID: 5IAI, 4Q0S, 4F2D) as an initial structure (Appendix A). Independent MD simulations (referred to as Run #1, #2, and #3, respectively) were performed for 100 ns for each initial structure. The MD simulation results are essentially the same; therefore, the following discussion is based on the results of Runs #1–#3. Figure 6 shows the results of MD simulations of ribitol (left, Run #1) and 1,3,5-pentanol (right) for each dihedral angle φ1 to φ4. The MD simulation results of Run #2 and #3 are shown in Appendix A.

MD simulations of 1,3,5-pentanol were also performed to investigate the role of OH2 and OH4 in ribitol (Figure 5, right). In 1,3,5-pentanol, φ1 and φ4 assume all three conformations (180°, +60°, −60°) almost equally (Appendix A). φ2 and φ3 also assume all three conformations, but −60° is slightly less populated for φ2 and +60° for φ3. Overall, transitions are frequent among conformers in 1,3,5-pentanol, while they are less so in ribitol. These results suggest that the presence of OH groups at positions 2 and 4 affects the frequency of conformational transitions. It is likely that this is because ribitol OH2 and OH4 form water-mediated or intramolecular hydrogen bonds that restructure the molecule.

## 3. Materials and Methods

### 3.1. Database Analysis of Ribitol and Ribitol Phosphate

Three-dimensional (3D) crystal structures of ribitol and ribitol phosphate (resolution less than 2.7 Å) were extracted from the Cambridge Crystallographic Data Centre (CCDC, as of May, 2020) and the Protein Data Bank (PDB, as of April 2020). Coordinates of ribitol phosphate were extracted that had at least one terminal CH_2_OH group. When analyzing tandem ribitol phosphate, the terminal ribitol with the CH_2_OH group was chosen. The dihedral angles of ribitol and ribitol phosphate are defined as φ1 (O1–C1–C2–C3), φ2 (C1–C2–C3–C4), φ3 (C2–C3–C4–C5), and φ4 (C3–C4–C5–O5).

### 3.2. Solution NMR Analysis

Ribitol was purchased from Tokyo Chemical Industry (Tokyo, Japan), and 1,3,5-pentanol was purchased from ChemCruz. Solution NMR analyses were performed using JNM-ECZ600R/S1, JNM-ECZ400S/L1 and EX270 spectrometers (JEOL, Tokyo, Japan). The probe temperature was set from 273 to 303 K. A sample (18–30 mg) was dissolved in 650 uL of D_2_O for signal assignment or 10 mM sodium acetate buffer, pH 6.0 (H_2_O:D_2_O = 1:1) for detection of OH chemical shifts. ^1^H and ^13^C chemical shits were reportedly related to the internal standard of 4,4-dimethyl-4-silapentane-1-sulfonic acid (DSS, 0 ppm). NMR chemical shifts of ribitol were assigned by analyzing 1D-^1^H, 1D-^13^C, 2D-DQF-COSY, NOESY, TOCSY, and ^1^H-^13^C HSQC spectra. NMR chemical shifts of ribitol phosphate were assigned by analyzing 1D-^1^H, 1D-^13^C, 2D-^1^H-^13^C HSQC, and 2D-^1^H-^13^C HMBC spectra. Stereospecific assignment of *pro*-*R* and *pro*-*S* protons in ribitol was conducted with the aid of 1D-^1^H-NMR spectral simulation by Mnova 14.1.1 (Mestrelab Research, Santiago, Spain). Due to the presence of a second-order term, the ^3^*J(*H,H) and ^2^*J(*H,H) coupling constants of ribitol were obtained by iterative comparison of simulated NMR spectra with observed NMR spectra obtained at ^1^H observation frequencies of 270, 400, and 600 MHz. ^3^*J(*C,H) coupling constants were obtained from HR-HMBC spectra [18]. A scaling factor (*n*) of 25 was used, and the digital resolutions for f1 (^13^C) and f2 (^1^H) were 4.4 and 0.7 Hz/point, respectively. The conversion from coupling constant ^3^*J*(H,H) to dihedral angle was done using the Haasnoot equation, which includes a correction for electronegativity of substituents and the orientation of each substituent relative to the coupled protons [20]. ^3^*J*(C,H) is also applied to an empirical prediction equation [21,22]. The exchange rate of hydroxyl protons with water was calculated from 2D chemical exchange spectroscopy [23,24]. Mixing times of 3 to 24 ms with increments of 3 ms were used. The exchange rate constant was calculated from the initial build-up rate of the diagonal peak volume over the exchange cross-peak. The temperature coefficient (ppb K^–1^) of hydroxyl protons in ribitol and 1,3,5-pentanol was measured by collecting a series of 1D-^1^H-NMR spectra at different temperatures (273, 278, 283, 288, 293, 298, and 303 K). NMR data processing was performed using Delta5. 3. 1 (JEOL, Tokyo, Japan), and NMR spectral analyses were performed using Mnova.

### 3.3. Solid-State ^13^C-CP-MAS NMR

^13^C CP-MAS spectra of crystalline ribitol (Tokyo Chemical Industry, Tokyo, Japan) were obtained using a Bruker Avance III 500 spectrometer at a ^1^H frequency of 500 MHz. A VPN probe (4 mm) was used at the spinning rate of 15,000 Hz or 8000 Hz with a spectral width of 300 ppm or 200 ppm, respectively. The data point was set to 2k. ^13^C chemical shifts were referenced to carboxyl carbon of glycine (176.03 ppm).

### 3.4. MD Simulation

MD simulations were performed using Discovery Studio 2019 [25]. The coordinates of ribitol (PDB ID: 5IAI, 4Q0S and 4F2D) were used as the initial structure (Appendix A), and the three simulations were performed independently. CHARMm36 was assigned as the force field [10]. Hydrogens were generated using the “Add Hydrogens” protocol in Discovery Studio. The simulation time was set to 100 ns. An explicit periodic boundary was used as the solvation model. An orthorhombic cell shape was used in the explicit periodic boundary solvation model. The minimum distance from the periodic boundary was set to 7.0 Å. For each ribitol coordinate, 182–184 water molecules were explicitly placed based on the protocol in Discovery Studio 2019. CHARMm36 was optimized using the TIP3P water model [26], which was used as the force field template. Explicit waters were 190 for 1,3,5-pentanol. Minimization of the initial coordinate was done in two steps. The first step is to eliminate distortion of the entire structure with the steepest descent algorithm. In the second step, minimization was performed with the adopted basis Newton–Raphson (NR) algorithm. Heating was done at 500 K. After equilibration, the time step was set to 2 fs, and production was done under *nPT* ensemble. MD simulations were also performed for 1,3,5-pentanol under the same protocol. The initial structure of 1,3,5-pentanol is a zigzag form with each dihedral angle of the carbon main chain being 180°.

## 4. Conclusions

Until the present study, it has been assumed that the C-C bonds of alditols are rather unstructured, and the molecule exhibits little or no stable folded structure. Here, NMR and MD simulation of ribitol reveal that ribitol is a dynamic conformational equilibrium of staggered conformations. Then, ribitol, is rather structured compared with model compound 1,3,5-pentanol. Conformational asymmetry may prevail in ribitol due to repulsion between OH2 and OH4 and to transient weak hydrogen bonds involving OH3. These findings provide a basis for understanding the dynamic structure of ribitol and the function of ribitol-related enzymes involved in ribitol biosynthesis and glycan chain elongation.

## Figures and Tables

**Figure 1 molecules-26-05471-f001:**
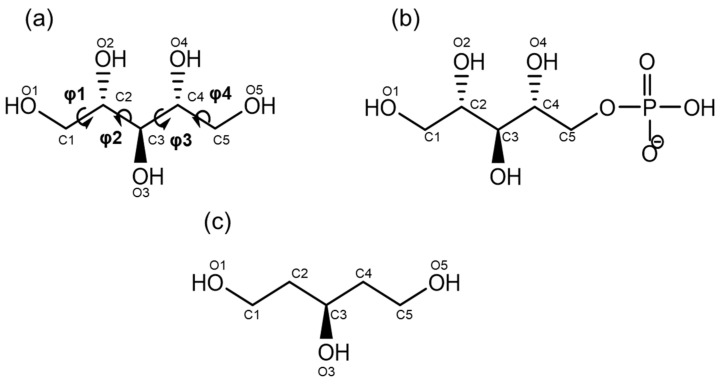
Chemical structures of the compounds examined in this study. (**a**) Ribitol, (**b**) ribitol phosphate (d-ribitol 5-phosphate is analyzed in this study, which is the form found in nature), and (**c**) 1,3,5-pentanol (reference compound). Carbon and oxygen numbering is indicated in the figure. The main-chain dihedral angles are defined as φ1 (O1–C1–C2–C3), φ2 (C1–C2–C3–C4), φ3 (C2–C3–C4–C5), and φ4 (C3–C4–C5–O5) throughout this manuscript.

**Figure 2 molecules-26-05471-f002:**
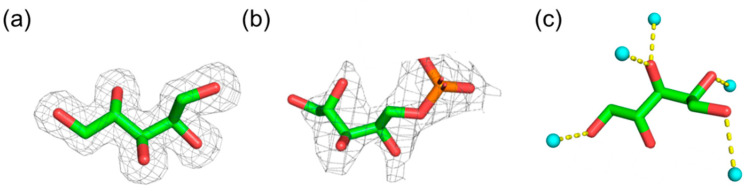
Representative 3D structure of ribitol and ribitol phosphate extracted from the database. (**a**) Crystal structure of ribitol (PDB ID: 5IAI). (**b**) Ribitol phosphate (PDB ID: 6H4F). (**c**) Ribitol interacting with water molecules (PDB ID: 4A0S). The structures are shown in stick representation. In (**a**,**b**), the electron density map is depicted in gray mesh contoured at 2.0 σ level. Proteins are omitted for clarity. In (**c**), water molecules hydrogen bonded with ribitol are shown in cyan sphere.

**Figure 3 molecules-26-05471-f003:**
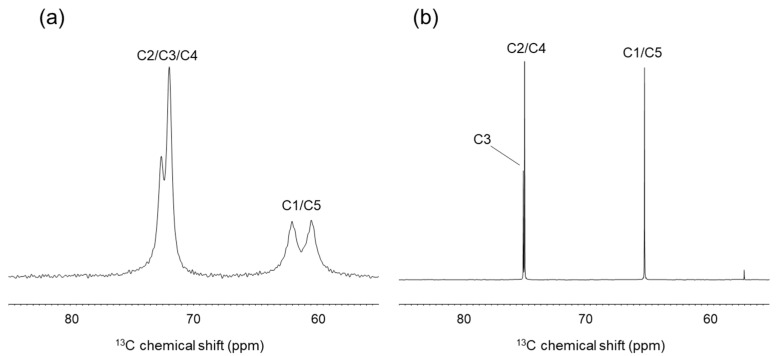
Comparison of solid-state and solution ^13^C-NMR spectra of ribitol. (**a**) Solid-state ^13^C CP-MAS -NMR spectrum of crystalline ribitol. (**b**) Solution ^13^C-NMR spectrum of ribitol dissolved in D_2_O.

**Figure 4 molecules-26-05471-f004:**
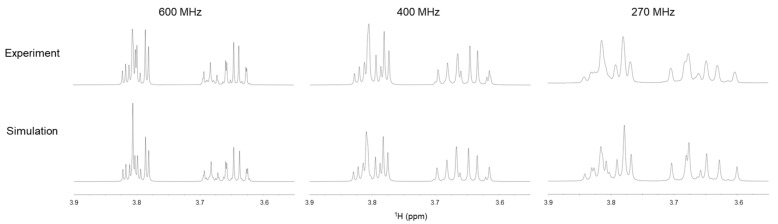
Solution ^1^H-NMR spectra of ribitol (**upper**) at ^1^H observation frequencies of 600 (**left**), 400 (**middle**), and 270 MHz (**right**). Simulated NMR spectra at each ^1^H frequency are shown in the lower panel.

**Figure 5 molecules-26-05471-f005:**
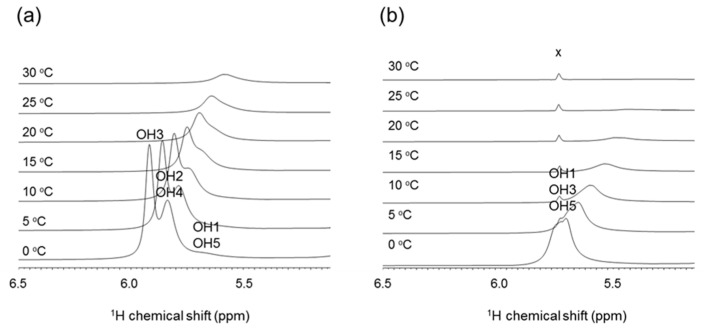
Part of 1D-^1^H-NMR (OH region) spectra of ribitol and 1,3,5-pentanol measured at 0, 5, 10, 15, 20, 25, and 30 °C in 10 mM sodium acetate buffer, pH 6.0 (H_2_O:D_2_O = 1:1). (**a**) Ribitol. (**b**) 1,3,5-Pentanol. x = impurity.

**Figure 6 molecules-26-05471-f006:**
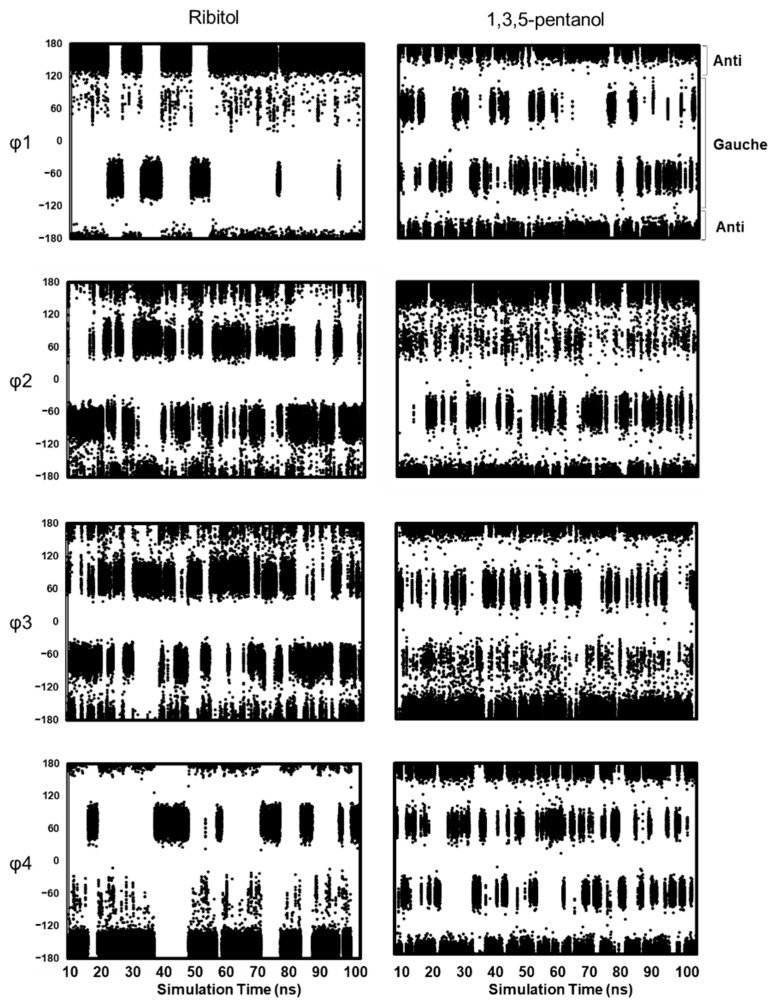
Dihedral angle distribution of ribitol (left, Run#1) and 1,3,5-pentanol (right) in MD simulations. The results of each dihedral angle are shown after 10 ns. The dihedral angles are defined as φ1 (O1–C1–C2–C3), φ2 (C1–C2–C3–C4), φ3 (C2–C3–C4–C5), and φ4 (C3–C4–C5–O5). φ1 and φ4 of ribitol are mostly distributed in anti (180°) and gauche (−60° for φ1 and +60° for φ4) (Appendix A). This is consistent with the NMR result, in which the populations of dihedral angle φ1 are estimated as 180°:−60°:60° = 64:36:0. For φ2 and φ3 of ribitol, gauche conformations (−60° and +60°) are dominant, but the anti-conformation is also populated. This is consistent with the NMR result (180°:−60°:+60° = 2:46:52). To validate the MD results quantitatively, ^3^*J*(H,H) and ^3^*J*(C,H) are calculated from the population of anti and gauche conformations, and the calculated *J* values are compared with experimental values (Table 3). The calculated *J* values are mostly consistent with the experimental values, suggesting that MD simulation quantitatively reflects the energy level of each conformer based on the Boltzmann distribution.

**Table 1 molecules-26-05471-t001:** Main chain dihedral angles (°), C1–C5 distance (Å), and resolution (Å) of ribitol and ribitol phosphate extracted from the CCDC and the PDB. NA; not available.

CCDC Deposition Number/DB ID	φ1 (O1–C1–C2–C3)	φ2 (C1–C2–C3–C4)	φ3 (C2–C3–C4–C5)	φ4 (C3–C4–C5–O5)	C1–C5 Distance	Resolution	Reference
ribitol							
1249410	171	−62	−172	71	4.5	-	[12]
662559	171	−61	−171	73	4.6	-	[13]
1015979	171	−61	−170	73	4.6	-	[14]
5IAI	173	166	60	170	4.5	1.6	NA
4Q0S	−176	−62	−167	176	4.5	1.9	[15]
4F2D	−150	−174	77	−147	4.5	2.3	NA
Ribitol phosphate							
6H4F	149	94	83	156	4.4	2.2	[16]
6H4M	131	−136	−143	−158	5.0	2.7	[16]
6HNQ	166	160	69	155	4.6	2.4	[16]
6KAM	−75	164	96	−168	4.8	2.5	[17]

**Table 2 molecules-26-05471-t002:** ^1^H and ^13^C chemical shift and coupling constant of ribitol determined in this study.

	Chemical Shift (ppm)
H1R(*pro-R*)	3.79
H1S(*pro-S*)	3.64
H2	3.81
H3	3.68
C1	65.1
C2	74.8
C3	74.9
	**Coupling constant (Hz)**
^3^*J*(H1R,H2)(*pro-R*)	3.00
^3^*J*(H1S,H2)(*pro-S*)	7.20
^3^*J*(H2,H3)	6.50
^2^*J*(H1R,H1S)	−12.70
^3^*J*(C1,H3)	3.8 *
^3^*J*(C3,H1S)	2.9 *

* ^3^*J*(C,H) may include errors due to the presence of strong ^3^*J*(H,H) coupling.

**Table 3 molecules-26-05471-t003:** Comparison of calculated and experimental coupling constants of ribitol. Calculated *J* values are obtained from the average of three independent MD simulations (mean ± SD).

	Calculated *J* Value (Hz)	Experimental *J* Value (Hz)
^3^*J*(H1R,H2)(*pro-R*)	3.11 ± 0.05	3.00
^3^*J*(H1S,H2)(*pro-S*)	7.57 ± 1.02	7.20
^3^*J*(H2,H3)	6.41 ± 0.09	6.50
^3^*J*(C1,H3)	4.20 ± 0.36	3.9
^3^*J*(C3,H1S)	2.67 ± 0.64	3.1

It seems that φ2 and φ3 tend to share the same dihedral angle, i.e., (φ2, φ3) = (+60°, +60°) or (−60°, −60°) during the MD trajectory. This differs from the database analysis, in which (φ2, φ3) mostly adopts (anti, gauche) or (gauche, anti). This discrepancy may come from the intermolecular contacts in the crystal lattice. The combination of φ2 and φ3 (+60°, +60° or −60°, −60°) can avoid steric hindrance between O2 and O4, and this arrangement seems reasonable.

## Data Availability

Not applicable.

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
