# Peer review of "Ribitol in Solution Is an Equilibrium of Asymmetric Conformations"

_molecules, 2021, doi:10.3390/molecules26185471_

Round 1

Reviewer 1 Report

This manuscript presents a conformational study of Ribitol. The relevance of this compound is clearly presented in the introduction (in terms of vitamin B2 and muscular dystrophy).  To the end of the study, the authors used database analysis, experimental NMR analysis and MD simulation. 

From my inspection of the manuscript, it could be accepted for publication in Molecules after 'very' major revision. My suggestions to improve the manuscript are, as follows:

(1) Introduction: Why is so important the knowledge of the Ribitol conformers behaviour? In the introduction you talk about the importance of Ribitol in terms of health, but you don´t clarify if any conformer is most reactive than other or if one of them is healthy but not the other(s). A similar comment can be presented for D-ribitol-5-phosphate: ¿is L-ribitol-5-phosphate stereoisomer less important than D-ribitol-5-phosphate?

(2) Figure 1: for the sake of clarification, include, at least, the number of the C and O atoms.

(3) Lines 108-111: I agree with the explanation, but ribitol and ribitol phosphate in the human body are not in crystalline form. Did you assess the solvent (water) interaction with ribitol? The ribitol behaviour in bloodstream is very different from the isolated crystal form.

(4) Line 157: Did you obtain the experimental and simulated spectra? Please include the spectra figures in the revised version to compare and evaluate your analysis.

(5) Lines 201-204: The sentence in these lines can be deleted as they are included in methodology.

(6) Lines 289-291: Can you give the %population of the main conformers analyzed (for example, based on a Boltzmann distribution)? Give the same information for 1,3,5-pentanol case.

(7) From Figure 6 it is not possible to conclude that OH3-O2 and OH3-O4 present hydrogen bonds! The typical H-O bond length is under 2 Å, but from the MD the most important population is over 3 Å. ¿How can you support that hydrogen bonding plays a significant role in ribitol conformations? Please include some potential references to support your discussion or, alternatively, delete this subsection of the manuscript.

(7) Line 400: The initial XYZ coordinates for MD simulations must be provided in supplementary materials.

(8) MD simulation is very poor described. There is only one reference in this subsection. How did you choose the number of water molecules? How were they situated in the solvation shell? Why did you choose the TIP3P as force field in the simulations? And so on...

Reviewer 2 Report

The manuscript gives a comprehensive overview of the conformational states of ribitol. The authors start with giving an overview of reported crystallographic structures, and then report on solid state and solution NMR experiments as well as molecular dynamics simulations. They conclude that in solution a dynamic equilibrium between asymmetric conformation occurs, which is in agreement with the experimental observations.

I would like to make the following comments:

  1. For the database analysis in table 1, the authors should be careful with the analysis of the crystal structures from the protein databank. The maximal resolution of 3 Å still allows for quite some conformational freedom, and is a global property. It is not uncommon that crystallographers of protein structures do not spend a whole lot of effort on the modelling of non-protein compounds, and it is conceivable that the reported data says more about the used force field in the model building than about the actual electron density. If possible, the authors may want to inspect the electron density around the ribitol.
  2. The authors use the Haasnoot equation (to which they refer as Karplus-like equation). This equation depends on a set of empirical constants, which should be reported in the work.
  3. The authors use this Haasnoot equation to determine the relative occurrence of different conformations, and subsequently roughly compare these distributions to the results of the molecular dynamics simulations. This is a comparison to secondary data, which can be avoided. Using the Haasnoot equation, the authors can easily calculate the J-values from the time series of the dihedral angles. They can then directly compare the ensemble averages of the J-values from the simulation to their experimentally measured counterparts, see e.g. [ https://doi.org/10.3390/ijms21010030 ].
  4. On page 5, the authors point out that the solution NMR data suggests a rapid transition between conformations. I agree that the transition must be fast, compared to the averaging time of the experiment. Can the authors quantify what ‘rapid’ means, or at least give an upper bound? Later on in the manuscript some estimates are given in the range of 193 s^-1. How does this relate to the transitions that are observed in the simulations of only 40 ns?
  5. While conformational transitions of the dihedral angles are observed in figure 5, these are not really frequent. This means it is difficult to estimate from the simulations an accurate transition state or the relative occurrences of the different conformations. The authors should consider to prolong the simulations further to obtain converged data.
  6. The authors determine the hydrogen bonds by monitoring the distances between the oxygen atoms. More elaborate geometric criteria are around to determine the occurrence of hydrogen bonds, that also include e.g. the Donor-hydrogen-acceptor angle. Why do the authors not use such a criterion to unambiguously show that the two hydrogen bonds O3 – O2 and O3 – O4 are mutually exclusive?

Round 2

Reviewer 1 Report

The manuscript has been revised in agreement with my comments. Therefore, I recommend to acept it in its actual form.